# Personality Profiles Associated with Long-Term Success in Bariatric Surgery: 24-Month Follow-Up

**DOI:** 10.3390/bs13100797

**Published:** 2023-09-26

**Authors:** Ignacio Montorio, María Izal, Ana Bellot, Javier Rodríguez, Mariano de Iceta

**Affiliations:** 1Psychology Faculty, Universidad Autónoma de Madrid, 28049 Madrid, Spain; ignacio.montorio@uam.es (I.M.);; 2Hospital Universitario Infanta Sofía, Universidad Europea de Madrid, 28702 Madrid, Spain

**Keywords:** obesity, bariatric surgery, successful personality profile

## Abstract

(1) Background: Bariatric surgery (BS) is highly effective for treating severe obesity in the long term. However, studies investigating predictors and personality profiles linked to BS success yield inconsistent results due to varying methodologies and limited research. This paper aims to identify personality profiles associated with BS success. (2) Method: The study involved 67 patients undergoing bariatric surgery, evaluated through clinical and personality measures. Weight loss was monitored at 6, 12, and 24 months post-surgery. Hierarchical case cluster analysis and iterative k-means cluster analysis identified distinct groups based on excess body mass index loss (%EBL) at these intervals. ANOVA was employed to compare personality profiles between groups. (3) Results: Average weight loss after 24 months was 67.2%. Two success profiles emerged: 46.5% showed very good success, achieving 90% EBL in 24 months, while 55% in the second cluster had less than 40% EBL throughout follow-up. The successful profile correlated with greater self-efficacy and improved emotional adjustment. (4) Conclusions: Successful BS outcomes were linked to personality traits promoting sustained weight loss post-surgery.

## 1. Introduction

Bariatric surgery (BS) is the most effective long-term treatment to ensure significant and sustained weight loss in people with morbid obesity and to improve obesity-related comorbidities and quality of life [1]. Although bariatric surgery can improve some of the health conditions related to obesity, such as diabetes, hypertension, and arthritis, the risk of surgical failure remains considerable. Thus, not all patients benefit from BS, with failure rates ranging from 4% to 53% [2]. Furthermore, up to 20–30% of patients undergoing bariatric surgery may eventually regain the full initial weight loss within 2 years of surgery [3]. Weight regains after bariatric surgery is perceived as an unexpected and unpleasant experience that contributes to hopelessness, embarrassment, and frustration that are partially motivated by internal and external circumstances such as psychosocial factors [4]. Given the high costs and potential risks associated with bariatric surgery, it is worth considering elements that will maximize its benefits over time. The success of BS requires a series of modifications in lifestyle, health, and eating habits that depend on the personality, coping, self-esteem, or psychopathology of the person undergoing surgery [5]. In this sense, the variable success of BS has been explained by a combination of basic characteristics of the patient (BMI, age, or sex) and previous psychosocial characteristics such as the person’s psychological profile and social conditions and their physiological, psychological, and behavioral responses to surgery, which can explain up to 60% of the variability [6].

Psychopathological symptomatology has been one of the most studied psychosocial aspects, as it is assumed that people with more psychological problems would have greater difficulty adapting to the demands of BS [7]. This hypothesis has been confirmed in several studies in which the presence of personality, eating, mood, or anxiety disorders was associated with poorer BS outcomes; on the contrary, the absence of severe psychiatric disorders was related to greater weight loss and improved quality of life after surgery [8,9,10]. However, there is also evidence that has not supported this relationship [11]. Given this lack of consistency of results, several authors have suggested ignoring the evaluation of psychopathology to focus on detecting potentially adaptive traits and strengths that lead to the success of BS [7,12]. 

Some studies have shown a relationship between the outcome of bariatric surgery and specific psychological factors such as personality characteristics [3]. Personality refers to the characteristics that change the patterns of thinking, feeling and behavior. In this regard, several papers have studied the predictive role of different personality traits on the success of surgery. In an interesting study using the NEO PI-R [13], a cluster analysis found two distinct groups following BS [14]. The first, characterized by low scores in neuroticism and high scores in openness, extraversion, agreeableness, and conscientiousness, was associated with the success of the surgery, whereas the second profile, opposite to the first, was associated with failure. The first profile meets the characteristics of a resilient personality [15]. These results are consistent with research using different personality measures such as the Temperament and Character Inventory TCI [16]. Thus, they found that trait cooperation, like agreeableness on the NEO-PI-R [16], was the main predictor of medium-term weight loss [8,17]. In addition, the review conducted by Claes and Muller [5] concluded that the traits of persistence, equivalent to conscientiousness [18], and self-direction, associated with high self-esteem and low neuroticism [19], contribute the most to the maintenance of weight loss. In contrast, harm avoidance, a concept related to neuroticism, is associated with BS failure [16]. Generali and De Panfilis [3] conducted a review of studies on the impact of preoperative personality traits on postoperative weight loss among individuals undergoing surgery for severe obesity with interesting findings. None of the studies they reviewed showed a relationship between pathological personality and variations in surgical success, while normal personality traits did show an impact on weight loss after bariatric surgery. They found that a personality pattern denoting a persistent ability to self-regulate despite impulses or the demands of the moment, as well as an ability to regulate impulses flexibly and seek help from others, emerges as a strong predictor of good outcomes in all studies.

However, there are inconsistent findings. Although most studies conclude that the novelty-seeking trait of the TCI, associated with impulsivity [17], is a negative predictor of BS, it has also been found to be positively associated with weight loss [20]. Other studies simply do not find an association between personality traits and BS success [10,21,22]. This inconsistency in results between studies could be due to the different methodologies used between studies: recruitment problems, resulting in the majority of participants in many of the studies being women; low uniformity in the methods used to assess both surgical outcomes and different personality traits, which may cause variation in the results [22]; and lack of control of relevant covariates, which poses significant challenges for reliable data interpretation [3].

In summary, it is difficult to reliably identify predictors of success in BS. This leads to uncertainty about the size, and even the directionality, of the relationship between psychosocial factors and success after BS, with a limited number of studies conducted in this regard [23]. The objective of this paper is to provide data from a longitudinal study on the association between personality and clinical factors and the success of BS to overcome previous inconsistencies. Unlike other studies, it will start from a person-centered model in which people who have undergone successful BS are previously identified, and we will subsequently characterize this group in terms of personality dimensions. For our research, we chose to use the Temperament and Character Inventory [24] to assess personality traits. This decision was based on the prevalence of its use in the existing literature, allowing us to draw more generalizable conclusions. We also sought to harmonize our personality constructs with those of the NEO PI-R [13], the most established and conventional personality assessment instrument.

## 2. Materials and Methods

### 2.1. Participants

The final sample was composed of 67 patients undergoing BS at the Infanta Sofía University Hospital in Madrid (Spain). The patients’ mean age was 48.5 years (SD = 9.7) with an age range of 24–67 years. Out of the sample, 65.7% were female and 34.3% were male. The average body mass index (BMI) before bariatric surgery was 41.5 (SD = 4.6) with a range of 33–52. All participants underwent an extensive multidisciplinary evaluation, which included a psychological evaluation. Therapeutic indications, requirements, and inclusion and exclusion criteria were based on the NIH Consensus Statement on Gastrointestinal Surgery for Severe Obesity [25]. The study was approved by the hospital’s research committee and met international ethical research standards (230731_181503, approval date: 13 May 2016). All patients signed informed consent prior to the operation.

### 2.2. Variables and Instruments 

The success of the BS corresponded to the percentage of excess body mass index loss (%EBL) which has proven to be an appropriate method for reporting weight loss [26,27]. This was calculated using the following formula:(preoperative Body Mass Index (BMI) − current BMI)/(preoperative BMI − 25) × 100)

The Trait-State Anxiety Inventory (STAI-T) is a widely used instrument to measure anxiety in individuals. Developed by Spielberger [28], it consists of two subscales: State Anxiety and Trait Anxiety. Each subscale contains 20 items, with a total of 40 items in the complete questionnaire. Participants rate the applicability of the statements on a 4-point Likert-type scale. The State Anxiety subscale measures anxiety experienced at a specific time, while the Trait Anxiety subscale assesses the general tendency to experience anxiety. A higher score means a higher level of anxiety. The STAI-T has consolidated reliability and validity data that are maintained in its Spanish version, in which it obtained a Cronbach’s alpha of 0.89 in the Trait Anxiety subscale and 0.84 in the State Anxiety subscale [28]).

The Bulimic Investigation Test, Edinburgh (BITE) [29] assesses the presence and severity of bulimia nervosa symptoms although it is not a diagnostic tool for bulimia nervosa. The BITE consists of 33 items divided into two subscales: the Severity subscale (severity of illness) and Symptom subscale (bulimic symptomatology). Participants respond to each item on a 6-point response scale. The total score is obtained by summing the item responses and can range from 0 to 96, indicating the severity of bulimic symptoms. Reliability data for the BITE were 0.62 for the Symptom subscale and 0.96 for the Severity subscale.

The Hamilton Rating Scale for Depression HRSD [30] is a clinical scale consisting of 17 items designed to measure different aspects of depressive symptomatology, such as depressed mood, loss of interest, fatigue, sleep disturbances, and somatic symptoms. Each item is scored on a scale of 0 to 4, and the scores are summed to obtain a total score that reflects the severity of depression. The scale has well-established psychometric properties (Cronbach’s alpha = 0.84) [31].

The Temperament and Character Inventory of Cloninger (TCI-R) is an instrument developed by Cloninger [24] to assess temperament and character traits in individuals, based on the personality theory developed by this author, which proposes that temperament refers to the biologically inherited and stable dimensions of personality, whereas character refers to the dimensions that are learned and develop throughout life. The TCI-R consists of 240 items, distributed in seven subscales that assess temperament dimensions (Novelty, Harm Avoidance, Reward Seeking, and Persistence) and character dimensions (Self-Direction, Cooperativeness, and Self-Transcendence). The instrument uses a 5-point Likert-type response scale and the Spanish language version has been validated, obtaining acceptable reliability coefficients in all subscales (Cronbach’s alpha between 0.65 and 0.85), similar to those of the original version [32].

### 2.3. Procedure

The laparoscopic gastric bypass is commonly utilized to treat morbid obesity, unless there were formal contraindications or patient-specific factors that made it inadvisable. Additional procedures, such as the tubular or vertical gastrectomy and single anastomosis duodeno-ileal bypass with sleeve gastrectomy (SADI-S), were employed to adapt surgical treatment for certain patient characteristics. Although surgical re-interventions currently approach 20%, the study participants had no previous bariatric surgeries or any during the follow-up period. Regular post-surgical follow-up is conducted until 18–24 months after bariatric surgery for dietary and nutritional monitoring, and with visits spaced 6 months or 1 year after this period by endocrinology. Likewise, the nursing staff conducts sessions for dietary re-education and verification of adaptation to the diet every two or three months. Nursing care is provided quarterly during the second year, every six months during the third, and annually thereafter. Mental health support is also offered during the first three years to assist with the recommendations made during the nutrition consultation, manage mental health issues, and assess the patient’s quality of life.

### 2.4. Statistical Analysis

The identification of differential profiles of bariatric success was carried out through cluster analysis, taking three indicators as the grouping variables: excess body mass index loss (%EBL) at 6 months, 12 months, and 24 months. Cluster analysis identified the natural groupings of bariatric success within the data sets that would not otherwise be obvious. According to Aldenderfer and Blashfield [33], five types of information are provided in cluster analysis: statistical package, cluster method, similarity of measures, procedure used to determine the number of groupings, and evidence of the validity of the groups. All analyses were conducted using the statistical program SPSS v26. Initially, hierarchical cluster analysis was performed, followed by a two-step cluster analysis. The hierarchical cluster analysis was performed using the Ward cluster method, which uses Z-scores to ensure an equal contribution to the classification and preservation of variance of the original subscale. Squared Euclidean distances were used as measures of the similarity of the cases. The two-step cluster analysis, which included the same indicators as the hierarchical analysis, allows automatic selection of the number of clusters by comparing the values of a chosen criterion model through different grouping solutions [34]. The final model of this study was selected through the Bayesian cluster criterion of Schwarz’s goodness of fit, based on a cohesion index using the silhouette coefficient. Although there is no universal recommendation for the appropriate sample size in a cluster analysis, a related methodology recommends a sample size of at least 2 k, where k specifies the number of cluster variables [35]. Therefore, the present study, with three cluster variables, meets the minimum criteria. Cluster analysis was followed by a comparison of means to identify significant differences between clusters, as well as to detect differences in clinical and personality variables. Finally, in order to identify the personality and emotional dimensions that contribute most to differentiating between the clusters derived from the previous analysis, a logistic regression analysis was performed using the enter method. The criterion variable was the identified clusters and the predictor variables were those that were statistically significant in the mean comparison analysis that was previously mentioned.

## 3. Results

Regarding the success of BS in the sample, at 6 months, %EBL was 55.2%; at 12 months, it was 71.2%; and at 24 months, it was 67.2%. Hierarchical cluster analysis suggested a two-cluster solution. When analyzing the fusion coefficients among cases and the dendrogram, it was found that the underlying structure of the data reflects a solution of two homogeneous groups without large fusion coefficients, which was confirmed by the automatic solution of the two-phase cluster analysis. The two extracted clusters corresponded to a group composed of individuals with clear BS success and a second heterogeneous group with poor bariatric success. We describe the evidence of the validity of the clusters, which is similar to that of Krug et al. [36]. Figure 1 contains information about the two-cluster solution. It can be observed how much each of the indicators contributes to forming the two clusters. The indicators of %EBL at 12 months and %EBL at 24 months were especially significant in the differentiation between clusters, with a somewhat lower contribution of %EBL at 6 months. The solution obtained had an appropriate silhouette index (0.7). The comparison of the distribution of the two clusters (N = 31, 55.5%; N = 36, 46.5%) yielded an adequate ratio of 1.16. Sánchez-Cabezudo and Larrad [27] propose that achieving a %EBL over 65% is an excellent result in BS, between 50% and 65% is considered good, and less than 50% is considered a failure. Figure 1 includes the percentage of people who are at the indicated levels of success for each indicator and cluster. It was observed that, in Cluster 1, the excellent category predominates in all %EBL indicators, whereas in Cluster 2, the failure category predominates. There were no significant differences in cluster membership based on gender, and there were also no significant gender differences in %EBL at any of the three assessment points. A comparison of the means between the two clusters for the three indicators %EBL showed statistically significant differences in all cases (*p* < 0.001) with statistically significant effect sizes according to the d Cohen index of 2.2 (95% CI [1.53, 2.8]) for %EBL at 6 months, 3.1 for %EBL at 12 months (95% CI [2.23, 3.8]), and 2.8 for %EBL at 24 months (95% CI [2.1, 3.6]). Figure 2 shows the means of %EBL in the three indicators for the two extracted clusters. It can be observed that, for all three indicators, the people of the bariatric failure cluster on average did not exceed a %EBL of 40%, which distances them from the usual criterion of general BS success of a %EBL of 50%. However, the success cluster showed values of 75% at 6 months and above 90% at 12 and 24 months.

Table 1 shows the comparison between the two empirical clusters of bariatric success for the clinical and personality variables relevant to this study. Comparisons between the clusters of bariatric success and failure yield statistically significant differences between the two, with higher scores in the successful group for the search for novelty, cooperation, and self-determination compared to the failure group. In turn, this latter group differed significantly from the former by its higher scores in harm avoidance, trait anxiety, and depression. Finally, logistic regression analysis was performed with the enter method, taking as a criterion variable the two clusters and the statistically significant variables of the previous analysis as predictors. A 79.1% correct classification (χ^2^; = 34, *p* < 0.001; Hosmer–Lemeshow = 8.06, *p* = 0.32; Cox and Snell = 0.4; Nagelkerke R^2^ = 0.53) was obtained. Only the variables Search for Novelty (Wald = 4.9, *p* < 0.005; Exp(B) = 0.949; 95% CI [0.90, 0.99]) and Self-Direction were statistically significant (Wald = 7.6, *p* < 0.005; Exp(B) = 0.95; 95% CI [0.92, 0.98]). Table 2 displays the outcomes for all variables integrated into the logistic regression analysis.

The effect size value is reported even when there are no significant differences between groups.

## 4. Discussion

The study shows an overall success of BS according to that found in most previous studies, as 70% of the patients exceed 50% %EBL [37]. The objective of this work was to provide prospective evidence that would help explain the association between personality and clinical factors and BS success. Unlike other studies predictive of the success of BS, this paper is based on grouping people as a function of success in weight loss at three different time points and subsequently characterizing the emerging profiles by personality and clinical traits. 

The cluster analysis performed, as suggested by the Silhouettes index obtained (0.70), allowed us to extract two differentiated groups: the first made up of people with clear success after BS and the second of participants for whom the surgery had limited results. Specifically, Cluster 1 was made up of people who, 24 months after BS, had an average %EBL of 90.4%, with all cases within the excellent range (higher than 65%) [27]. On the other hand, in Cluster 2, the mean %EBL was 39.1%, which indicates that the differentiation established between the two profiles distinguishes those who reached the highest level of success from those for whom BS had failed or had only obtained slight success. 

Regarding the differences between the two clusters in the clinical and personality variables, it was observed that the people belonging to Cluster 2 scored significantly higher in Harm Avoidance, Trait Anxiety, and Depression. This is consistent with previous works that have studied the relationship between psychosocial variables and BS success [5,14]. As for Cluster 1, the cooperative personality characteristic appears in this study as a factor significantly associated with BS success, consistent with as the findings of Generali and De Panfilis [3] in their recent review on the influence of personality traits on post bariatric surgery weight loss. This result supports the hypothesis proposed by Goodpaster [38] or Sullivan and Cloninger [17], suggesting that people with higher scores in Cooperation and Agreeableness may have better postsurgical results due both to their greater adherence to follow-up programs, which are usually conducted in groups, and to their greater social support network, which would favor a better quality of life. The patients of Cluster 1 also obtained significantly higher scores in the dimensions of Novelty-Seeking and Self-Direction. Specifically, these two personality characteristics were presented as those with greater predictive relevance because they could classify four out of five participants in one of the clusters. Regarding Self-Direction, Van Hout et al. [7] concluded in their review that it is one of the most consolidated variables in the literature to predict weight loss after BS. High scores in this dimension imply the individual has strong cognitive control, self-efficacy, and the ability to pursue meaningful goals [24]. People who obtain higher scores in this dimension are more likely to exhibit a greater capacity for maintaining diligent behaviors, such as consistently adhering to dietary and clinical monitoring appointments, even when faced with occasional lack of rewards or immediate gratification from counterproductive actions [3].

As for Novelty Seeking, it was also found in Gordon et al. [20] as the main predictor. However, this trait, associated in some studies with impulsivity and pathological eating habits [39], has also been shown to be a predictor of low success for weight loss after BS [17]. In their study, Gordon et al. [20] hypothesized that the positive association between Novelty Seeking and weight loss was due to the development of eating patterns such as purging. However, this hypothesis can be ruled out in the present study, which included purging evaluation, in which the scores between the two clusters did not differ. As an alternative hypothesis, it is proposed to stop interpreting the role of the Novelty Seeking trait in isolation and instead consider its interaction with the rest of the traits and variables. Thus, Novelty Seeking is also linked by Cloninger [24] to exploratory behavior, that is, to a motivational factor or an incentive activator that can modify behaviors [40]. From this point of view, higher Novelty Seeking could imply a greater predisposition to initiate new behaviors, but the type of behavior that encourages such initiation would depend on the rest of the associated traits and context. For example, Leombruni et al. [41] found that Novelty Seeking paradoxically predicts thinness when associated with high perfectionism. Claes and Muller [5] analyzed several studies in which, when high BAS (impulsivity, sensation-seeking) and high BIS (anxiety, neuroticism) were presented together, more binge eating and worse surgery outcomes were predicted. This would suggest that Novelty Seeking hinders the success of BS when combined with traits such as neuroticism, as observed in Cluster 2 of this study. 

The possible limitations of this study include, firstly the small sample size and the limited follow-up of only two years, since the review by Spirou et al. [42] recommends a minimum follow-up of 36 months to adequately allow for psychological and physical adjustment following bariatric surgery. Secondly, the only criterion of success was weight loss, without considering a possible increase in quality of life or decrease in psychopathological symptomatology. Third, the effect of covariates that, according to previous studies might have a significant moderating role in the outcomes of bariatric surgery such as self-efficacy, coping strategies, locus of control, or gender of the participants [42], has not been controlled for. Fourth, previous studies have identified personality disturbances as significant predictors of BS success, so it would be desirable for future studies, in addition to assessing patients’ personality traits, to use standardized interviews to diagnose personality disorders and thus differentiate between the predictive role of functional and dysfunctional personality patterns [3]. Fifthly, although there are no gender differences in the main results of the study, the sample was two-thirds female, which should be taken into account when assessing the generalizability of the results. Finally, although bulimia nervosa is an important disorder associated with morbid obesity [43], the choice of the BITE scale (a scale for the detection and description of binge eating) is not a sufficiently sensitive instrument to assess the risk of all eating disorders that may increase after BS. From an application perspective, although the presence of any psychological pattern should not be considered as an inclusion–exclusion criterion for BS, it does seem of interest to promote behavioral characteristics of success before and after BS. National practice guidelines suggest the use of a formal psychosocial assessment in patients presenting with symptoms of psychopathology [44,45]. This recommendation has been supported by other widely cited reviews published over the past two decades [46,47]. Regarding the assessment of patients’ personality traits, although temperamental traits might have a constitutional basis, they are not unmodifiable; clarifying the processes and behaviors through which these traits contribute to successful postsurgical weight loss will identify potential areas for intervention [3]. In this sense, the results of this work point to the potential benefit of practicing self-efficacy behaviors in patients undergoing BS, which has been recently conducted in the field of eating disorder interventions [48]. In a broader sense, recent applications of the concept of human agency, referring to the ability to influence the functioning and course of events through one’s own actions [49], have great potential for facilitating healthy behaviors.

## 5. Conclusions

Several biopsychosocial factors, such as personality and emotional characteristics, influence the maintenance of weight loss after costly bariatric surgery. This study has identified a personality and emotional profile characterized by harm avoidance, anxiety, and depression as predictors of treatment failure 24 months post-BS. On the other hand, traits such as cooperativeness, agreeableness, novelty-seeking, and self-directedness comprise a personality profile that significantly benefits from bariatric surgery at 6-, 12-, and 24 months post-surgery. In particular, the combination of the latter two traits has significant predictive value for success in bariatric surgery. Motivation and predisposition to initiate new health-promoting behaviors in individuals with strong cognitive control, self-efficacy, and goal-directed behavior are factors that contribute to success after BS. These traits align with recent advancements in Bandura’s human agency model [49] and may have therapeutic applications. Agency is a central concept in psychotherapy that refers to clients’ ability to attribute thoughts, emotions, and actions to themselves, and to take initiative and responsibility for their behavior in daily life [50]). 

## Figures and Tables

**Figure 1 behavsci-13-00797-f001:**
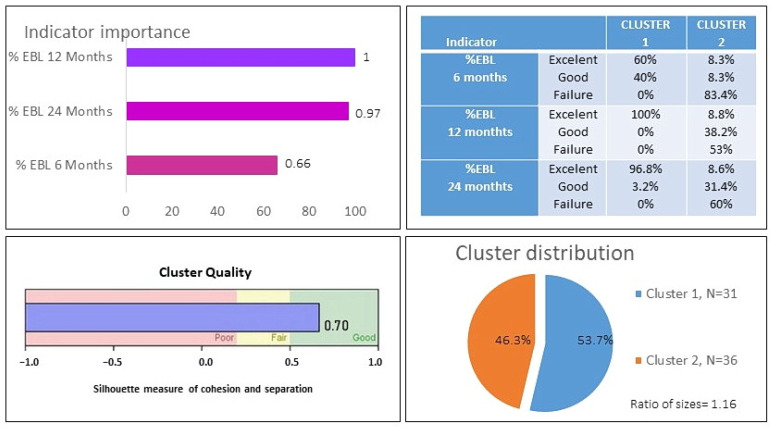
Results of the clustering procedure based on success in bariatric surgery.

**Figure 2 behavsci-13-00797-f002:**
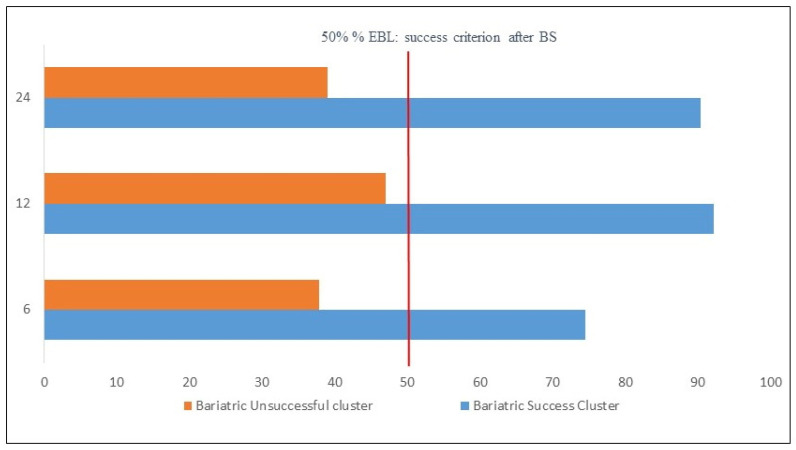
Differences in excess percent BMI loss (%EBL) in the three indicators for Successful Bariatric Surgery Cluster and Unsuccessful Bariatric Surgery Cluster.

**Table 1 behavsci-13-00797-t001:** Personality and clinical characteristics in Successful and Unsuccessful Bariatric Surgery groups.

	Successful Bariatric Cluster (N = 31) M (SD)	Unsuccessful Bariatric Cluster (N = 36) M (SD)	t	Cohen’s *d* [95% CI]
Harm Avoidance	90.7 (19.8)	107.2 (20.1)	3.3 **	0.83 [0.91, 1.98]
Novelty Seeking	100.5 (12.8)	89.6 (14.8)	3.2 **	1.44 [0.33, 1.32]
Reward Dependence	107.4 (22.9)	104.5 (17.0)	n.s.	−0.14 [−0.66, 0.33]
Persistence	113.9 (20.2)	112.4 (29.3)	n.s.	−0.06 [−0.53, 0.42]
Cooperation	144.8 (12.2)	137.4 (15.1)	2.1 *	0.53 [0.1, 1.02]
Self-Direction	155.2 (19.0)	131.9 (23.5)	4.4 **	1.09 [0.6, 1.61]
Self-Transcendence	61.2 (12.9)	67.2 (17.9)	n.s.	−0.28 [−0.15, 0.86]
Trait Anxiety	18.8 (10.3)	26.6 (13.0)	2.6 *	0.66 [0.2, 1.1]
Depression	1.1 (2.4)	3.3 (4.5)	2.3 *	0.68 [0.2, 1.17]
Purging	11.6 (9.5)	12.8 (8.9)	n.s.	0.13 [−0.35, 0.61]

** *p* < 0.001. * *p* < 0.05.

**Table 2 behavsci-13-00797-t002:** Predicting Successful and Unsuccessful Bariatric Surgery Cluster membership by logistic regression.

	Wald	*p*	Exp(B)	95% C.I.
Harm Avoidance	1.591	0.207	1.022	0.98–1.05
Novelty Seeking	4.50	0.032	0.949	0.90–0.99
Cooperation	0.05	0.822	0.994	0.946–1.04
Self-Direction	7.6	0.006	0.951	0.92–0.98
Trait Anxiety	0.647	0.421	1.028	0.96–1.01
Depression	0.867	0.352	1.120	0.88–1.42

## Data Availability

The data presented in this study are available on request from the corresponding author. The data are not publicly available due to ethical restrictions.

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
