# Peer review of "Personality Profiles Associated with Long-Term Success in Bariatric Surgery: 24-Month Follow-Up"

_behavsci, 2023, doi:10.3390/bs13100797_

Round 1
Reviewer 1 Report
Dear authors, this manuscript is a good contribution to the field of bariatric surgery. However, there are some limitations that you have to address.
1. About the reference style. Please review the authors' guidelines because is necessary to put them in the correct way, not in APA style.
2. Introduction. Lines 40-41 are not clear, please review them.
3. Materials and methods. Please complete the data of the ethical committee with the number or code assigned to your research.
4. About the use of BITE Test. Is this tool enough to assess the risk of all eating disorders (ED)? We currently know that people living with obesity and people after BS could increase the risk of ED, in this case, BITE is specific for bulimic behavior. Please explain how this impacts your results.
5. Results. The first thing that must be in this section is the description of the participants. Please do a new table with the data, for example: sex, age, gender, BMI previous surgery, and more that could be important to understand the results.
6. Please, can you add the complete information about how you classify participants in three levels of BS success.
7. Please add a new table with the data obtained after the logistic regression analysis. P-value for each variable in the model.
8. Please organize your table and figure along with the text not at the end of the results section.
Please, review the English grammar because there are some misspellings in the manuscript.
Author Response
We appreciate the reviewers' comments as they improve the manuscript.
Response to comment 1.
The references in the text have been revised and modified according to the journal's guidelines.
Response to comment 2
The sentence on lines 40-41 has been modified to make it easier to understand in relation to the following sentence detailing the meaning (marked in red).
Response to comment 3
The requested information on ethics has been included with the approval code and date (marked in red).
Response to comment 4
In line with the reviewer's observation that BITE may be an insufficient instrument to detect risk for eating disorders as a whole, it has been included as a limitation in the discussion section (marked in red).
Response to comment 5
With regard to including a table with the sample data, we believe that this would not be sufficiently large to form a table in the text of the article. The research protocol, in accordance with the ethical limitations imposed, has prevented us from having access to more complete information on the patients. Additional information available on the sample has been expanded in the text in the participants section.
Response to comment 6
Larrad and Sanchez-Cabezudo (2004) reviewed the quality indicators in bariatric surgery and criteria for long-term success. They propose three levels of success based on the percentage of excess body mass index loss (%EBL). They point out that since the percentage of excess BMI lost correlates strongly with the percentage of excess body mass index lost (PSP), individualised assessment with a single parameter (%EBL) can be proposed: excellent result if above 65%, good if between 50 and 65% and failure if below 50%. This information is included on page 5 (marked in red).
Response to comment 7.
A table (table 2) has been added with the data obtained from the logistic regression indicating the p-value for each variable in the established model.
Response to comment 8
The placement of the tables and figures has been done by the publisher itself, not by the authors, so we maintain the current placement and leave the final arrangement to the editors' consideration.
Reviewer 2 Report
Would you please report the method(s) of BS applied the post-surgical condition and treatment partly depent on them (methods section, results, if meaningful; discussion).
Author Response
We appreciate the reviewers' comments as they improve the manuscript.
A procedure section has been incorporated in methods, which extensively describes the surgical procedures used and the follow-up for post-surgical monitoring (marked in red).
Reviewer 3 Report
The publication submitted to me for review analyses issues that are currently significant and relevant.
I consider that the article is well written, however it has certain shortcomings that need to be corrected:
- the most important point - lack of conclusions - this should be corrected;
- in the cited literature, it is worth adding more recent publications;
- the literature is missing source 36 and 43.
Author Response
We appreciate the reviewers' comments as they improve the manuscript.
Response to comment 1.
A specific and extensive section on conclusions has been added (marked in red).
Response to comment 2.
Prior to writing the manuscript, a careful review of the literature was carried out, including this year. In fact, the first citation in the article is from the year 2023. The literature on personality and bariatric surgery is relatively scarce so we chose to consider it irrespective of the year of publication. Many of the citations that can be considered old refer to authorship of assessment instruments, consensus criteria on the success of bariatric surgery or quality criteria for the use of statistical techniques. All these references remain valid. Even so, just over 40% of the citations are from the last 10 years. In the last 5 years there is hardly any original research, although some reviews have been included in the manuscript.
Response to comment 3.
Regarding "the literature is missing source 36 and 43" it is an error in the edition that has already been corrected by deleting those lines.
Reviewer 4 Report
Personality Profiles associated with long-term success in bariatric surgery: 24-month follow-up
Review: Bariatric surgery is one approach to dealing with severe obesity. Past research examining personality correlates of successful surgery has been inconsistent. The present study seeks to examine the association between personality characteristics and bariatric surgery success over three time points in a sample of 67 patients. Cluster analysis was used to identify distinct weight groups and compare them for differences in personality/temperament characteristics.
Comments
· Introduction: The authors did a good job of reviewing the literature regarding bariatric surgery and personality correlates. One thing that I think may help the reader is why did the authors pick the specific psych/personality constructs they chose. There is some justification in the intro, but I think it would need to be spelled out at the end. “Based on the past literature and… we chose the following constructs for our person-centered analyses.”
· Materials and Methods: The authors included a sample of 67 participants and have data from three separate time points. This type of data is very difficult to acquire and time-consuming and produces a unique sample. My question is – are there any issues with power, particularly with the cluster analysis? Additionally, the sample is majority women? Does this potentially affect generalizability?
· Results: The statistics were well-described and appear to be appropriate for the questions. In the table with Cohen’s d, I would probably just add the cohen’s d for the null models. It’ll show the extent of the null effect (i.e., how small it was). Also, because the Cohen’s d was reported, I would add a note denoting the size of the effect.
Author Response
We appreciate the reviewers' comments as they improve the manuscript.
Response to comment 1
The reviewer's interesting remark about the choice of personality constructs based on TCI (Cloninger) has been resolved by including - according to the reviewer - a paragraph at the end of the introduction justifying it. In this paragraph the importance of comparability with previous results and generalisability is pointed out. It also indicates that an effort has been made in the introduction and discussion to compare the results in terms of NEO-PI.
Response to comment 2
a) Power
Researchers can follow different guidelines to choose the right algorithms and determine what constitutes a convincing cluster grouping. To the best of our knowledge, there are no clearly established ways to calculate a priori statistical power for cluster analysis. Dalmaijer et al. (2022) opted for a simulation approach to estimate power and classification accuracy by systematically varying the cluster size, the number of clusters, the number of different features between clusters, the effect size within each different feature, and the cluster covariance structure in the generated datasets. They then subjected these datasets to the various dimensionality reduction approaches (none, multidimensional scaling or approximation and uniform multiple projection) and clustering algorithms (k-means, hierarchical agglomerative clustering with Ward linkage and Euclidean distance, etc.). In addition, they simulated additional datasets to explore the effect of sample size and cluster separation on statistical power and classification accuracy. They found that clustering results were driven by large effect sizes or the accumulation of many smaller effects between characteristics, and were mostly unaffected by differences in covariance structure. They point out that sufficient statistical power can be achieved with relatively small samples (N= 20-30 per subgroup) as long as the group separation found is sufficiently large. Also, there might be a loss of power to detect a difference between clusters when there is no homogeneity between subjects in the same cluster compared to that expected from a random sample.
In this sense, in our study the separation between clusters is large, the minimum sample size is met and there is homogeneity in each cluster.
Dalmaijer ES, Nord CL, Astle DE. Statistical power for cluster analysis. BMC Bioinformatics. 2022 May 31;23(1):205. doi: 10.1186/s12859-022-04675-1. PMID: 35641905; PMCID: PMC9158113.
b) Gender
With regard to gender differences, the sample is indeed composed of a larger number of women. This is in line with the bariatric literature as there is a female bias in most studies and this fact should be considered with regard to the generalisability of the studies. Statistical analyses have been performed to compare whether the main results of this study are compromised by gender. The results show that there are no statistically significant differences by gender with respect to more females in either cluster, nor are there significant differences by gender with respect to %EBL at each of the three assessment time points (6, 12 and 24 months). This evidence has been added to the results section. A brief comment has also been added to the limitations section.
Response to comment 3
Cohen's d has been included for all models and the requested note has been added (table 1)
Round 2
Reviewer 1 Report
Dear authors, your manuscript has been improved but there are some minors comments that need to resolve prior aceptation.
1. Please in the statistical analysis section you have to add the complete analysis done it, such as logistic regression analysis.
2. Please in table number 2 add the OR, is not enough with the Wald coefficient. Also, the figure and tables from the results section have to be in the correct way after their aparance in the text.
3. Please, put the reference in the correct way in the text, in this journal is necessary to pun in a number not like in the APA.
---
Author Response
1. Please in the statistical analysis section you have to add the complete analysis done it, such as logistic regression analysis.
The statistical analysis section has been completed
2. Please in table number 2 add the OR, is not enough with the Wald coefficient. Also, the figure and tables from the results section have to be in the correct way after their aparance in the text.
Table 2 includes the Exp (B) whose value is the odd-ratio
Figures and tables in the results section have been placed in the correct place (as far as possible).
3. Please, put the reference in the correct way in the text, in this journal is necessary to pun in a number not like in the APA.
References have been correctly placed in the text according to editorial standards (numbered).